DOI: 10.1038/s41467-018-07482-6　　**OPEN**

# Efficient and stable sky-blue delayed fluorescence organic light-emitting diodes with CIE$_y$ below 0.4

Chin-Yiu Chan[1], Masaki Tanaka[1], Hajime Nakanotani[1] & Chihaya Adachi[1,2]

Organic light-emitting diodes utilizing thermally activated delayed fluorescence is a potential solution for achieving stable blue devices. Sky-blue devices (CIE$_y$ < 0.4) with high stability and high external quantum efficiency (>15%) at 1000 cd m$^{-2}$ based on either delayed fluorescence or phosphorescence are still limited and very hard to achieve simultaneously. Here, we report the design and synthesis of a new thermally activated delayed fluorescence emitter, **3Ph$_2$CzCzBN**. A sky-blue device based on **3Ph$_2$CzCzBN** exhibits a high external quantum efficiency of 16.6% at 1000 cd m$^{-2}$. The device shows a sky-blue electroluminescence of 482 nm and achieves Commission Internationale de l' Eclairage coordinates of (0.17, 0.36). The sky-blue device exhibits a superb LT$_{90}$ of 38 h. This is the first demonstration of high-efficiency and stable sky-blue devices (CIE$_y$ < 0.4) based on delayed fluorescence, which represents an important advance in the field of blue organic light-emitting diode technology.

[1] Center for Organic Photonics and Electronics Research (OPERA), Kyushu University, 744 Motooka, Nishi, Fukuoka 819-0395, Japan. [2] International Institute for Carbon Neutral Energy Research (WPI-I2CNER), Kyushu University, 744 Motooka, Nishi, Fukuoka 819-0395, Japan. Correspondence and requests for materials should be addressed to C.A. (email: adachi@cstf.kyushu-u.ac.jp)

Organic light-emitting diodes (OLEDs) are attractive technology for display and lighting due to their high flexibility, high efficiency and light-weight[1–5]. In order to commercialize OLEDs for various electronic applications, the color purity[6–11], efficiency[12–15], driving voltage[16–19] and device stability[20-27] of OLEDs are some of the important parameters to be considered.

Blue and sky-blue phosphorescent OLEDs (PhOLEDs) with $CIE_y < 0.4$ have been developing over 15 years[28], and the resultant PhOLEDs with high efficiencies and long operation times are still difficult to achieve simultaneously at high brightness[17,18,20–22,28–30]. Regarding to the device stability, the LT$_{90}$s (90% of the initial luminance of 1000 cd m$^{-2}$) of blue and sky-blue PhOLEDs are usually less than 10 h, which are insufficient for commercialization[22,29,30]. In 2014, Forrest and coworkers reported an improved stability of a single-unit blue phosphorescent OLEDs (dopant: Ir (dmp)$_3$) with the use of a graded-dopant-concentration technique in the emitting layer (EML)[20]. The graded device achieved a moderate external quantum efficiency (EQE) of 9.5% at 1000 cd m$^{-2}$ with a driving voltage of 7.7 V. Such devices exhibited a LT$_{90}$ of ~40 h at an initial luminance of 1000 cd m$^{-2}$. Recently, in 2017, Forrest and coworkers further extended the device lifetime by introducing an extra molecule functioning as a hot excited-state manager in the EML of the graded devices[21]. The managed device achieved a similar EQE of 9.6% with a higher driving voltage of 9.0 V at 1000 cd m$^{-2}$, and resulted in a lengthened LT$_{90}$ of ~140 h. Nonetheless, the EQEs of both graded and managed devices were still less than 10% at 1000 cd m$^{-2}$, rendering them less competitive with the commercial fluorescent blue emitters. Moreover, the driving voltages for both devices at 1000 cd m$^{-2}$ were around 8 V or above, which were rather high for practical use.

Meanwhile, efficient OLEDs based on thermally activated delayed fluorescence (TADF) were first demonstrated by Adachi and coworkers in 2012[31]. TADF is believed to be a potential solution to achieve stable blue OLEDs with an internal quantum efficiency (IQE) of 100%. Since then, there are numerous reports on efficient blue and sky-blue TADF OLEDs with $CIE_y < 0.4$[32-34]. However, reports on stable blue and sky-blue TADF OLEDs are still rare and limited[23–25]. Again, the LT$_{90}$s (90% of the initial luminance of 1000 cd m$^{-2}$) of blue and sky-blue TADF OLEDs are always less than 10 h[23,25]. Recently, Adachi and coworkers reported a stable sky-blue TADF OLED by the use of a n-type host material (SF3-TRZ)[24]. The sky-blue TADF OLED was able to achieve a EQE of 7% with a driving voltage of 5.7 V at 1000 cd m$^{-2}$. The corresponding device exhibited a comparable LT$_{90}$ of ~20 h (at an initial luminance of 1000 cd m$^{-2}$) to those reported sky-blue PhOLEDs[20,21]. However, same as PhOLEDs[20,21], the EQE of the stable sky-blue TADF OLED was still below 10%. As a result, there is still a room for designing new TADF emitters to concurrently achieve blue or sky-blue OLEDs with a high EQE and a high stability at high brightness.

Here we show that a sky-blue device based on a new fluorescence emitter (3Ph$_2$CzCzBN) exhibits a high external quantum efficiency of 16.6% at 1000 cd m$^{-2}$. The device shows a sky-blue electroluminescence of 482 nm and achieves Commission Internationale de l' Eclairage coordinates of (0.17, 0.36). The sky-blue device exhibits a superb LT$_{90}$ of 38 h. This is the first demonstration of high-efficiency (>15%) and stable sky-blue devices (CIE$_y$ < 0.4) based on delayed fluorescence at 1000 cd m$^{-2}$.

## Results
**Molecular design and synthesis**. The long-lived triplet excitons are believed to be lethal to the device stability for both phosphorescent and TADF OLEDs, because hot triplet excitons are

generated by triplet-triplet annihilation or triplet-polaron annihilation processes[35–37]. Therefore, decreasing the delayed lifetimes of the blue TADF emitters is useful to enhance the device stability. Benzonitrile is one of the promising building blocks for constructing stable blue TADF emitters[23]. However, blue benzonitrile-based TADF emitters usually possess long delayed lifetimes, thus resulting in severe rolloffs and short device lifetimes in OLEDs[23]. Recently, it is proven that the introduction of two different donor units in the TADF emitters is effective to lower the energy of the high-lying localized triplet excited state ($^3$LE), in which a good mixing of the $^3$LE state with the lowest charge transfer triplet excited state ($^3$CT) can facilitate a faster reverse intersystem crossing rate ($k_{RISC}$)[38]. Moreover, such faster $k_{RISC}$ results in a shorter delayed lifetime of the TADF emitters, which eventually improves the device stability since triplet exciton intensity in devices will be reduced. Herein, we synthesized a new blue benzonitrile-based TADF emitter, 3Ph$_2$CzCzBN, in two steps (Fig. 1). 4CzBN was also synthesized in one step as a reference[23]. Both compounds were well characterized by $^1$H and $^{13}$C NMR, mass spectrometry analysis and elemental analysis (Supplementary Figures 1–7). Their purities were also confirmed by high-performance liquid chromatography (Supplementary Figures 8 and 9). It is expected that the hetero-donor-based TADF emitter (3Ph$_2$CzCzBN) should show a shorter delayed lifetime and a longer device lifetime than that of the homo-donor-based reference compound (4CzBN). Quantum-chemical calculations based on 3Ph$_2$CzCzBN and 4CzBN were first performed by time-dependent density functional theory (TD-DFT) at B3LYP/6–31 G(d) level. The calculated highest occupied molecular orbital (HOMO), lowest unoccupied molecular orbital (LUMO), singlet (S$_1$) and triplet (T$_1$) energy levels, oscillator strength ($f$) and $\Delta E_{ST}$s were depicted in Fig. 1. From the TD-DFT calculation, 3Ph$_2$CzCzBN and 4CzBN showed similar $\Delta E_{ST}$s of 0.13 and 0.14 eV and high oscillator strengths ($f$s) of 0.0619 and 0.0818, respectively. The calculated HOMOs of both compounds were mainly localized on the carbazole donors with some extensions on the benzonitrile acceptor, whereas the LUMOs are fully localized on the benzonitrile acceptor.

**Photophysical, electrochemical and thermal properties**. The photophysical properties of 3Ph$_2$CzCzBN and 4CzBN were first examined in toluene solution at the concentration of 10$^{-5}$ M. Both compounds showed absorption bands from 290 to 450 nm, in which the absorption bands at around 400–450 nm were assigned as the intramolecular charge transfer (ICT) bands (Fig. 2). As expected, by the introduction of 3,6-diphenylcarbazole, 3Ph$_2$CzCzBN showed a redshifted absorption than that of 4CzBN. The energy gaps of 3Ph$_2$CzCzBN and 4CzBN were calculated from their onset values of 10% of the absorption maxima and were found to be 2.79 and 2.89 eV, respectively. Upon photo-excitation at the ICT band, 3Ph$_2$CzCzBN and 4CzBN both showed structureless emissions in the blue region with the peak maxima of 464 and 443 nm, respectively (Fig. 2 and Table 1). The redshifted absorption and emission spectrum of 3Ph$_2$CzCzBN were consistent to the TD-DFT calculation. The T$_1$s of 3Ph$_2$CzCzBN and 4CzBN were found to be 2.60 and 2.66 eV, which were estimated from the phosphorescence spectra in toluene solutions at 77 K (Fig. 2 and Table 1). The $\Delta E_{ST}$s of 3Ph$_2$CzCzBN and 4CzBN were calculated to be 0.19 and 0.23 eV, respectively. The photoluminescence quantum yields (PLQYs) of 3Ph$_2$CzCzBN and 4CzBN were found to be 16 and 9%, respectively, before degassing. After purging N$_2$ into the solutions, the corresponding PLQYs of 3Ph$_2$CzCzBN and 4CzBN were greatly enhanced and found to be 95 and 63% (Table 1). Doping 15 wt% of 3Ph$_2$CzCzBN and 4CzBN in 3,3'-di(9H-carbazol-9-yl)−1,1'-

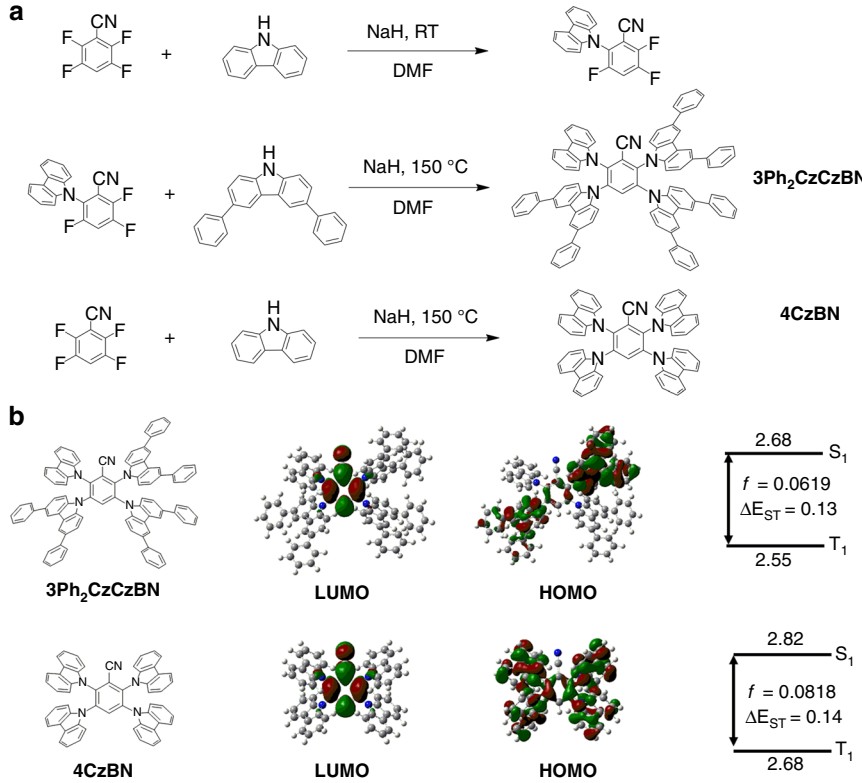

**Fig. 1** Synthetic scheme and quantum chemical calculations of the blue emitters. **a** Synthetic scheme of **3Ph₂CzCzBN** and **4CzBN**. **b** Quantum chemical calculations of **3Ph₂CzCzBN** and **4CzBN**

**Table 1 Photophysical and thermal properties of 3Ph₂CzCzBN and 4CzBN**

| Compound | $\lambda_{Abs}$ (nm)[a] | $\lambda_{em}$ (nm)[a,b] | PLQY (%)[a,c] | PLQY (%)[c,d,f] | $\tau_p$ (ns)[a] | $\tau_d$ (μs)[a] | $S_1$ (eV)[g] | $T_1$ (eV)[h] | $\Delta E_{ST}$ (eV) | HOMO (eV)[i] | LUMO (eV)[j] | $T_d$ (°C)[k] |
|---|---|---|---|---|---|---|---|---|---|---|---|---|
| **3Ph₂CzCzBN** | 347,422 | 464 | 95[d]/16[e] | 91 | 3.7 | 10 | 2.79 | 2.60 | 0.19 | −5.69 | −2.78 | >500 |
| **4CzBN** | 334,403 | 443 | 63[d]/9[e] | 51 | 1.6 | 50 | 2.89 | 2.66 | 0.23 | −5.79 | −2.75 | 455 |

[a]$1 \times 10^{-5}$ M in toluene.
[b]Emission maximum at room temperature in toluene
[c]Absolute photoluminescence quantum yield (PLQY) by an integrating sphere
[d]Degassed/purged by argon
[e]Aerated
[f]15% wt doped in mCBP
[g]Determined by absorption edge in toluene
[h]Determined by emission maximum in toluene at 77 K
[i]Determined by cyclic voltammetry, where HOMO = $-(E_{ox} + 4.8)$ eV
[j]LUMO = $-(E_{red} + 4.8)$ eV
[k]Decomposition temperature determined by 5 wt% loss

biphenyl (mCBP) resulted in slightly lower PLQYs of 91 and 51% (Supplementary Table 1).

The transient decay profiles of **3Ph₂CzCzBN** and **4CzBN** were first studied in degassed toluene solution. The prompt lifetimes of **3Ph₂CzCzBN** and **4CzBN** were found to be 3.7 and 1.6 ns, whereas the delayed lifetimes were found to be 10 and 50 μs, respectively (Supplementary Figure 10). Notably, the delayed lifetime of **3Ph₂CzCzBN** was greatly shortened comparing to that of **4CzBN**, which may be ascribed to its smaller $\Delta E_{ST}$. The TADF properties of **3Ph₂CzCzBN** and **4CzBN** in the thin films were also studied by the streak camera. Simply doping 15 wt% of **3Ph₂CzCzBN** or **4CzBN** in mCBP, the delayed lifetimes at 300 K were found to be 5.2 and 13.7 μs (Supplementary Figure 11), which were consistent to that in the solution state. The TADF behaviors were confirmed by performing temperature-dependent transient decay study of the 15 wt% doped films from 50 to 300 K (Fig. 2 and Supplementary Figures 11, 12 and 14). The detailed rate constants were also calculated and tabulated in Supplementary Table 1. The activation energies ($\Delta E_a^{TADF}$) for the delayed

fluorescence were also calculated from the Arrhenius plots of $k_{RISC}$ vs. $1/T$ based on the relationship $k_{RISC} = \exp(-\Delta E_a^{TADF}/k_B T)$, where $k_{RISC}$ is the rate constant of reverse intersystem crossing, $k_B$ is the Boltzmann's constant and T is the temperature. The $\Delta E_a^{TADF}$s of **3Ph₂CzCzBN** and **4CzBN** were found to be 42 and 100 meV, respectively (Supplementary Table 2 and 3, Supplementary Figures 13 and 15). Also, the $k_{RISC}$s of **3Ph₂CzCzBN** and **4CzBN** were found to be $4.44 \times 10^5$ and $2.15 \times 10^5$ s$^{-1}$, respectively (Supplementary Table 1). The smaller $\Delta E_a^{TADF}$ of **3Ph₂CzCzBN** may account for the faster $k_{RISC}$.

The electrochemical properties of **3Ph₂CzCzBN** and **4CzBN** were determined by cyclic voltammetry in N,N-dimethylformamide solutions (Supplementary Figures 16–19). The HOMOs of **3Ph₂CzCzBN** and **4CzBN** were determined to be −5.69 and −5.79 eV, respectively, whereas their corresponding LUMOs were found to be −2.78 and −2.75 eV. The HOMOs of **3Ph₂CzCzBN** and **4CzBN** were also estimated by measuring their neat films with the photoelectron spectrometer. Their HOMOs were found to be −5.91 and −5.99 eV, respectively (Supplementary Figures 20

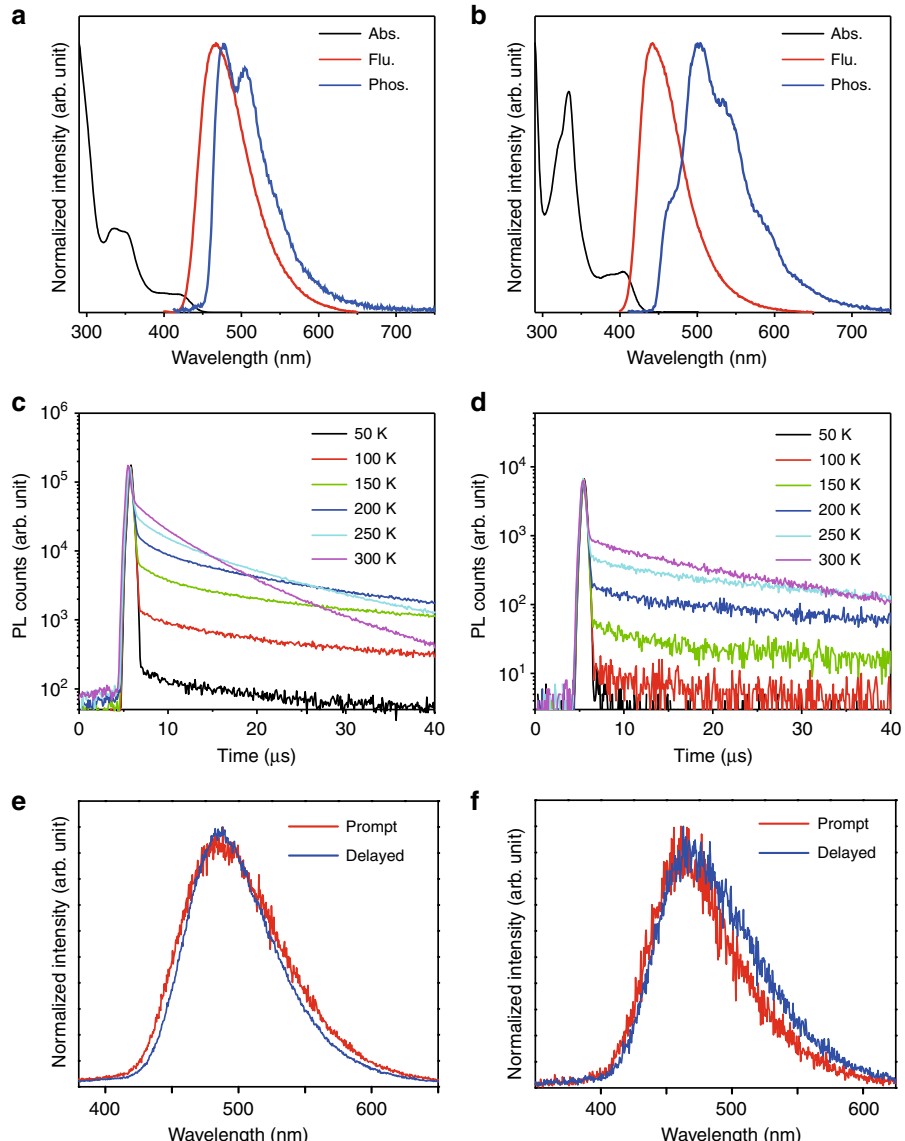

**Fig. 2** Photophysical properties of **3Ph$_2$CzCzBN** and **4CzBN**. **a** Absorption, fluorescence and phosphorescence spectra of **3Ph$_2$CzCzBN** in toluene. **b** Absorption, fluorescence and phosphorescence spectra of **4CzBN** in toluene. **c** Temperature-dependence transient decay profiles of a thin film of 15 wt% of **3Ph$_2$CzCzBN** doped in mCBP. **d** Temperature-dependence transient decay profiles of a thin film of 15 wt% of **4CzBN** doped in mCBP. **e** Prompt and delayed spectra of a thin film of 15 wt% of **3Ph$_2$CzCzBN** doped in mCBP. **f** Prompt and delayed spectra of a thin film of 15 wt% of **4CzBN** doped in mCBP

and 21). Furthermore, the thermogravimetry - differential thermal analysis (TG-DTA) and differential scanning calorimetry (DSC) analysis were also performed to examine the thermal properties of **3Ph$_2$CzCzBN** and **4CzBN**. From TG-DTA measurement, **3Ph$_2$CzCzBN** and **4CzBN** showed excellent thermal stability with T$_d$s (5% weight loss) of > 500 and 455 °C, respectively (Supplementary Figures 22 and 23). The melting points of the sublimed samples were found to be 433 and 448 °C, respectively (Supplementary Figures 22 and 23). From DSC measurement, **3Ph$_2$CzCzBN** showed glass transition temperature (T$_g$) at 240 °C (Supplementary Figure 24). The DSC measurement of **4CzBN** was not performed, since its melting point is close to its T$_d$. All the photophysical, electrochemical and thermal properties were tabulated in Table 1 and Supplementary Table 1.

**Device characterization and performance**. To evaluate the electroluminescence (EL) properties of **3Ph$_2$CzCzBN** and **4CzBN**, multilayered OLEDs (device **A** and **B**) were fabricated using the

following structure: Indium–tin oxide (ITO)-coated glass (100 nm)/ HATCN (10 nm)/ Tris-PCz (30 nm)/ mCBP (5 nm)/ 15 wt% of **3Ph$_2$CzCzBN** (device **A**) or 15 wt% of **4CzBN** (device **B**): mCBP (30 nm)/ T2T (10 nm)/ BPy-TP2 (40 nm)/ LiF (0.8 nm)/ Al (100 nm), in which hexaazatriphenylenehexacarbonitrile (HATCN) was the hole-injection layer, 9,9',9"-triphenyl-9H,9'H,9"H–2,3':6',2"-tercarbazole (Tris-PCz) was the hole-transporting layer, 3,3'-di(9H-carbazol-9-yl)-1,1'-biphenyl (mCBP) was the electron-blocking layer, 15 wt% of **3Ph$_2$CzCzBN** (device **A**) or 15 wt% of **4CzBN** (device **B**) doped in mCBP was the emitting layer (EML), 2,4,6-tris(biphenyl-3-yl)–1,3,5-triazine (T2T) was the hole-blocking layer, 2,7-di(2,2'-bipyridin-5-yl)triphenylene (BPy-TP2) was the electron-transporting layer and LiF and Al were the electron injection layer and cathode, respectively. The materials used and the device structure for devices **A** and **B** were depicted in Supplementary Figures 25 and 26.

The detailed OLEDs performance for devices **A** and **B** were shown in Supplementary Figures 27–34 and Table 2. Both devices **A** and **B** showed sky-blue emission with maximum peaks at 482

and 477 nm, respectively. The CIE coordinates of device **A** and **B** were found to be (0.17, 0.37) and (0.19, 0.32), respectively. Device **A** showed a maximum EQE of 15.9% and dropped to 15.0% at 1000 cd m$^{-2}$, whereas device **B** showed a severe rolloff with a maximum EQE of 9.7% and a much lower EQE of only 5.8% at 1000 cd m$^{-2}$. The driving voltages of devices **A** and **B** were found to be 6.3 and 7.6 V at 1000 cd m$^{-2}$, respectively. Devices **A** and **B** were driven at a constant DC current at room temperature for testing the device stability. The LT$_{90}$s of devices **A** and **B** were found to be 4 and 0.1 h, respectively. Hole-only and electron-only devices (HOD and EOD) based on the structures of devices **A** and **B** were also fabricated to understand more about the carrier transport properties (Supplementary Figure 35). It was found that both devices showed similar carrier transport properties. For **3Ph$_2$CzCzBN**, the current density ratios of HOD to EOD at 6 and 7 V were 20 and 57, respectively. Meanwhile, for **4CzBN**, the current density ratios of HOD to EOD at 6 and 7 V were 14 and 47, respectively. It seemed that the carrier transport properties in device **A** and **B** were not the major determining factor for the stability enhancement. In fact, we believed the enhanced stability of device **A** was originated from the shortening of the delayed lifetime of **3Ph$_2$CzCzBN** (5.2 μs) than that of **4CzBN** (13.7 μs). The shortening of the delayed lifetime of **3Ph$_2$CzCzBN** lowered the triplet exciton intensity in EML, in which the rolloff in device **A** was significantly reduced. Moreover, the diminished triplet exciton intensity also avoided exciton-exciton annihilation and exciton-polaron annihilation, thus enhancing the device stability.

To further improve the device stability, devices **C** and **D** were fabricated with the following device structure: Indium–tin oxide (ITO)-coated glass (100 nm)/ HATCN (10 nm)/ Tris-PCz (30 nm)/ mCBP (5 nm)/ 15 wt% of **3Ph$_2$CzCzBN** (device **C**) or 20 wt % of **3Ph$_2$CzCzBN** (device **D**): mCBP (30 nm)/ SF3-TRZ (10 nm)/30 wt% of Liq: SF3-TRZ (50 nm)/ Liq (2 nm)/ Al (100 nm), in which HATCN was the hole-injection layer, Tris-PCz was the hole-transporting layer, mCBP was the electron-blocking layer, 15 wt% of **3Ph$_2$CzCzBN** (device **C**) or 20 wt% of **3Ph$_2$CzCzBN** (device **D**) doped in mCBP was the emitting layer (EML), 2-(9,9'-spirobi[fluoren]−6-yl)−4,6-diphenyl-1,3,5-triazine (SF3-TRZ) was the hole-blocking layer, 30 wt% of 8-quinolinolato lithium (Liq): SF3-TRZ was the electron-transporting layer and Liq and Al were the electron injection layer and cathode.

Comparing to device **A**, device **C** showed the identical EL peak maximum of 482 nm with the CIE coordinates of (0.18, 0.37). Device **C** also showed a same maximum EQE of 15.9%. The turn-on voltage of device **C** was 3.7 V, which was lower than that of device **A** (4.0 V). Furthermore, the driving voltage of device **C** was found to be 6.4 V at 1000 cd m$^{-2}$. Although device **C** showed a slightly lower EQE of 14.3% at 1000 cd m$^{-2}$, the LT$_{90}$ of device **C** was significantly improved by four times to 16 h. Further optimizing the device performance, the doping concentration of **3Ph$_2$CzCzBN** was increased from 15 wt% (device **C**) to 20 wt% (device **D**). In device **D**, the EL peak maximum was mildly redshifted to 486 nm with CIE coordinates of (0.18, 0.39). Device **D** showed a maximum EQE of 17.9%. Interestingly, device **D** showed an insignificant rolloff and achieved a EQE of 17.2% at 1000 cd m$^{-2}$. Meanwhile, a driving voltage of 5.8 V was obtained. Most importantly, the LT$_{90}$ of device **D** was lengthened to 32 h, which is a 320-times increase comparing to that of the **4CzBN**-based device. The detailed device characteristics of device **C** and **D** were depicted in Fig. 3 and tabulated in Table 2. Additionally, color tuning based on device **D** for achieving bluer emission is possible by optimizing the thicknesses of each layer or by the use of micro-cavity effect and/or filters.

An optical simulation based on device **D** has been performed to study the effect of the thickness of the ETL (10 to 60 nm) on the EL emission color (Supplementary Figure 36). It was found

that the EL emission would be blue-shifted with the thinner ETL layer. However, too thin ETL may result in imbalanced carrier mobilities of the device. Therefore, an attempt on reducing the thickness of the ETL from 50 nm (device **D**) to 20 nm (device **E**) was performed. Device **E** resulted in comparable EQEs of 17.8% and 16.6% at the maximum value and 1000 cd m$^{-2}$, respectively. Also, device **E** showed a slightly lengthened stability with a LT$_{90}$ of 38 h. Most importantly, device **E** achieved better CIE coordinates of (0.17, 0.36) at 1000 cd m$^{-2}$, which is bluer than that of (0.18, 0.39) in device **D** (Table 2 and Supplementary Figures 37–40).

A summary of the best reported blue and sky-blue OLEDs based on delayed fluorescence and phosphorescence was listed in Supplementary Table 4. To our best knowledge, the result present here is the first report of a sky-blue OLED (CIE$_y$ < 0.4) with a high external quantum efficiency (EQE~16.6%) and a long stability (LT$_{90}$ ~38 h) at high brightness (1000 cd m$^{-2}$), which represents a big milestone to the field of the blue OLED technology.

## Discussion

A new TADF emitter, **3Ph$_2$CzCzBN**, has been synthesized and characterized. It showed a higher PLQY and a shorter delayed lifetime than that of the reference compound, **4CzBN**. Devices based on **3Ph$_2$CzCzBN** showed a sky-blue EL of 482 nm and CIE coordinates of (0.17,0.36). Such device achieved a high EQE of 16.6% at 1000 cd m$^{-2}$. Most importantly, **3Ph$_2$CzCzBN**-based TADF OLEDs showed superior device stability. With an initial luminance of 1000 cd m$^{-2}$, an exceptional LT$_{90}$ of 38 h was achieved. In conclusion, a stable sky-blue OLED (CIE$_y$ < 0.4) and a high external quantum efficiency (>15%) at high brightness (1000 cd m$^{-2}$) were achieved simultaneously with the use of a newly synthesized TADF emitter, **3Ph$_2$CzCzBN**. This outstanding result definitely suggests the possibility for achieving efficient and stable blue OLEDs by TADF technology.

## Methods

**General**. All reagents were used as received from commercial sources and were used without further purification. Chromatographic separations were carried out using silica gel (200–300 nm). The two materials investigated in this paper were synthesized by following the procedures described below. All compounds were purified twice by temperature gradient vacuum sublimation. $^1$H, $^{13}$C and $^{19}$F nuclear magnetic resonance (NMR) spectra were obtained in CDCl$_3$ or acetone-d$_6$ with a Bruker Biospin Avance-III 500 NMR spectrometer at ambient temperature. Chemical shifts (δ) are given in parts per million (ppm) relative to tetramethylsilane (TMS; δ = 0) as the internal reference. Mass spectra were measured in positive-ion atmospheric-pressure chemical ionization (APCI) mode on a Waters 3100 mass detector. Elemental analyses (C, H and N) were carried out with a Yanaco MT-5 elemental analyzer.

**Synthesis of 3Ph$_2$CzCzBN**. Under nitrogen atmosphere, 3,6-diphenyl-9H-carbazole (957 mg, 3 mmol) was dissolved in dry N,N-dimethylformamide (30 mL) in a two-neck round-bottom flask equipped with a condenser. The reaction mixture was cooled to 0 °C, then NaH (120 mg, 3 mmol) was added. The reaction mixture was slowly warmed to room temperature and stirred for half an hour. After that, 2-(9H-Carbazol-9-yl)-3,5,6-trifluorobenzonitrile (322 mg, 1 mmol) was added and the reaction was heated to 150 °C for 16 h. The reaction was quenched with water and the precipitate was filtered off. The crude product was purified by column chromatography. Yield: 976 mg (80%). $^1$H NMR (500 MHz, Acetone-d$_6$, 298 K, relative to Me$_4$Si): δ = 9.07 (s, 1 H), 8.36 (s, 2 H), 8.29 (d, 4 H, 10.0 Hz), 7.95–8.00 (m, 10 H), 7.55–7.70 (m, 18 H), 7.30–7.45 (m, 20 H), 7.18 (t, 2 H, 7.0 Hz). $^{13}$C NMR (126 MHz, Acetone-d$_6$): δ = 143.1,143.1, 142.1, 142.0, 141.6, 141.5, 141.0, 140.7, 140.2, 140.0, 136.2, 135.9, 135.8, 130.6, 130.5, 128.8, 128.8, 128.7, 128.5, 128.4, 127.6, 126.8, 126.7, 126.7, 126.5, 126.42, 125.8, 122.9, 122.0, 120.5, 120.4, 120.4, 113.4, 112.9, 112.7, 112.7.MS (APCI) calcd. for C$_{91}$H$_{57}$N$_5$: m/z = 1220.5; found: 1221.0 [M]$^+$. Elemental analysis calcd. (%) for C$_{91}$H$_{57}$N$_5$: C 89.55, H 4.71, N 5.74; found: C 89.51, H 4.65, N 5.72.

**Synthesis of 4CzBN**. Under nitrogen atmosphere, 9H-carbazole (668 mg, 4 mmol) was dissolved in dry N,N-dimethylformamide (30 mL) in a two-neck round-bottom flask equipped with a condenser. The reaction mixture was cooled to 0 °C, then

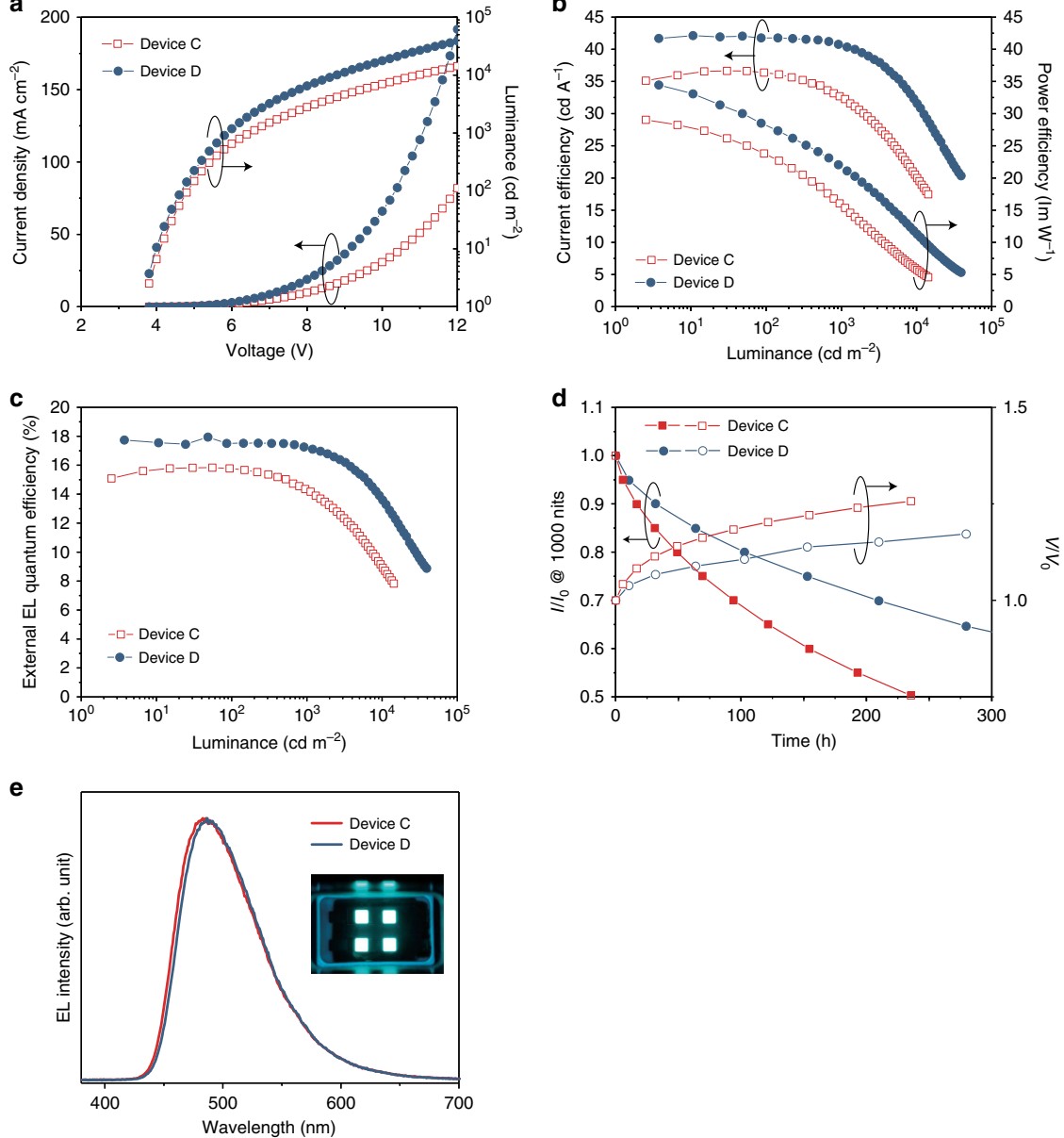

**Fig. 3** Device characteristics of device **C** and **D**. **a** Current density and luminance versus voltage. **b** Power efficiency and current efficiency versus luminance. **c** External quantum efficiency versus luminance. **d** Luminance and voltage versus time (at an initial luminance of 1000 cd m$^{-2}$). Device structure: ITO (100 nm)/ HATCN (10 nm)/ Tris-PCz (30 nm)/ mCBP (5 nm)/ 15 wt% of **3Ph2CzCzBN** (device **C**) or 20 wt% of **3Ph2CzCzBN** (device **D**): mCBP (30 nm)/ SF3-TRZ (10 nm)/ 30 wt% of Liq:SF3-TRZ (50 nm)/ Liq (2 nm)/ Al (100 nm). **e** EL spectra of device **C** and **D** and photo of the resulting OLED

NaH (160 mg, 4 mmol) was added. The reaction mixture was slowly warmed to room temperature and stirred for half an hour. After that, 2,3,5,6-tetra-fluorobenzonitrile (175 mg, 1 mmol) was added and the reaction was heated to 150 °C for 16 h. The reaction was quenched with water and the precipitate was filtered off. The crude product was purified by column chromatography. Yield: 557 mg (73%). $^1$H NMR (500 MHz, CDCl$_3$, 298 K, relative to Me$_4$Si): δ = 8.43 (s, 1 H), 7.77–7.79(m, 8 H), 7.35 (d, 4 H, 8.0 Hz), 7.30–7.32 (m, 4 H), 7.18 (t, 4 H, 7.5 Hz), 7.09–7.17 (m, 12 H). $^{13}$C NMR (126 MHz, CDCl$_3$): δ = 139.2, 138.8, 137.8, 136.9, 136.3, 125.7, 124.3, 123.9, 121.3, 121.0, 120.4, 120.2, 118.3, 113.0, 109.9, 109.2. MS (APCI) calcd. for C$_{55}$H$_{33}$N$_5$: $m/z$ = 763.3; found: 763.3 [M]$^+$. Elemental analysis calcd. (%) for C$_{55}$H$_{33}$N$_5$: C 86.48, H 4.35, N 9.17; found: C 86.50, H 4.29, N 9.18.

**Quantum chemical calculation**. All calculations were carried out using the Gaussian 09 program package. The geometries in the ground state were optimized via TD-DFT calculations at the B3LYP/6-31 G(d) level. TD-DFT calculations for the $S_0 \rightarrow S_1$ and $S_0 \rightarrow T_1$ transitions using the B3LYP functional were then performed according to the optimized geometries of the lowest-lying singlet and triplet states, respectively.

**Photophysical measurements**. Toluene solutions containing these two materials (10$^{-5}$M) were prepared to investigate their absorption and photoluminescence characteristics in the solution state. Thin-film samples (15% wt- doped in mCBP 100 nm) were deposited on quartz glass substrates by vacuum evaporation to study their exciton confinement properties in the film state. Ultraviolet–visible absorption (UV-vis) and photoluminescence (PL) spectra were recorded on a Perkin-Elmer Lambda 950 KPA spectrophotometer and a JobinYvon FluoroMax-3 fluorospectrometer. Phosphorescent spectra were recorded on a JASCO FP-6500 fluorescence spectrophotometer at 77 K. Absolute PL quantum yields were measured on a Quantaurus-QY measurement system (C11347-11, Hamamatsu Photonics) under nitrogen flow and all samples were excited at 360 nm. The transient PL decay characteristics of solution samples were recorded using a Quantaurus-Tau fluorescence lifetime measurement system (C11367-03, Hamamatsu Photonics). The prompt and delayed PL spectra of the samples were measured under a vacuum using a streak camera system (Hamamatsu Photonics, C4334) equipped with a cryostat (Iwatani, GASESCRT-006-2000, Japan). A nitrogen gas laser (Lasertechnik Berlin, MNL200) with an excitation wavelength of 337 nm was used.

**Table 2 Device performance of OLEDs based on 3Ph₂CzCzBN and 4CzBN**

| Device | $\lambda_{EL}$ (nm)[a] | V (V)[b] | EQE (%)[c] | PE (lm W$^{-1}$)[d] | CE (cd A$^{-1}$)[e] | CIE (x,y)[f] | LT (h)[g] |
|---|---|---|---|---|---|---|---|
| A | 482 | 4.0/5.0/6.3 | 15.9/15.8/15.0 | 27.8/22.7/17.0 | 36.3/36.0/33.9 | (0.17,0.37) | 4/15 |
| B | 477 | 4.0/5.3/7.6 | 9.7/8.7/5.8 | 20.9/11.3/4.9 | 26.6/20.0/12.1 | (0.19,0.32) | 0.1/0.4 |
| C | 482 | 3.7/4.8/6.4 | 15.9/15.8/14.3 | 29.2/23.5/15.7 | 36.6/36.2/32.6 | (0.18,0.37) | 16/47 |
| D | 486 | 3.8/4.6/5.8 | 17.9/17.6/17.2 | 34.5/28.1/21.8 | 41.7/41.7/40.9 | (0.18,0.39) | 32/103 |
| E | 482 | 4.0/4.5/5.5 | 17.8/17.6/16.6 | 33.9/29.1/20.7 | 39.2/38.9/36.1 | (0.17,0.36) | 38/118 |

[a]Emission peak maximum at 1000 cd m$^{-2}$.
[b]Voltage at onset, 100 cd m$^{-2}$ and 1000 cd m$^{-2}$
[c]External Quantum Efficiency: maximum, value at 100 cd m$^{-2}$, value at 1000 cd m$^{-2}$
[d]Power Efficiency: maximum, value at 100 cd m$^{-2}$, value at 1000 cd m$^{-2}$
[e]Current Efficiency: maximum, value at 100 cd m$^{-2}$, value at 1000 cd m$^{-2}$
[f]At 5 mA cm$^{-2}$
[g]Lifetime (at an initial luminance of 1000 cd m$^{-2}$): LT$_{90}$ / LT$_{80}$

**Thermal properties**. Thermal gravimetry-differential thermal analysis (TG-DTA) was performed by Bruker TG-DTA 2400SA with a heating rate of 10 °C min$^{-1}$ under nitrogen atmosphere. Differential scanning calorimetry (DSC) analysis was performed by Netzsch DSC204 Phoenix calorimeter at a scanning rate of 5 °C min$^{-1}$ under N$_2$ atmosphere.

**Cyclic voltammetry measurements**. Cyclic voltammetry (CV) was carried out on a CHI600 voltammetric analyzer at room temperature with a conventional three-electrode configuration consisting of a platinum disk working electrode, a platinum wire auxiliary electrode and an Ag wire pseudo-reference electrode with ferrocenium–ferrocene (Fc$^+$/Fc) as the internal standard. Argon-purged N,N-dimethylformamide was used as solvent for scanning the oxidation with tetra-butylammonium hexafluorophosphate (TBAPF$_6$) (0.1 M) as the supporting electrolyte. The cyclic voltammograms were obtained at a scan rate of 100 mV s$^{-1}$.

**Device fabrication and measurements**. The OLEDs were fabricated through vacuum deposition of the materials at ca. $2.0 \times 10^{-4}$ Pa onto indium–tin-oxide-coated glass substrates having a sheet resistance of ca. 15 Ω per square. The indium–tin oxide surface was cleaned ultrasonically and sequentially with acetone, isopropanol and deionized water, then dried in an oven, and finally exposed to ultraviolet light and ozone for about 10 min. Organic layers and aluminum were deposited at a rate of 1–2 Å/s. Subsequently, LiF and Liq were deposited at 0.1–0.2 Å/s. The devices were exposed once to nitrogen gas after the formation of the organic layers to allow the fixing of a metal mask to define the cathode area. After fabrication, the devices were immediately encapsulated with glass lids using epoxy glue in nitrogen-filled glove boxes (O$_2$–0.1ppm, H$_2$O-0.1ppm). For all OLEDs, the emitting areas were determined by the overlap of two electrodes as 0.04 cm$^2$. The J-V-luminance characteristics were evaluated using a Keithley 2400 source meter and an absolute external quantum efficiency (EQE) measurement system (C9920-12, Hamamatsu Photonics, Japan). Device operational stability was measured using a luminance meter (CS-2000, Konica Minolta, Japan) at a constant DC current at room temperature.

**Optical simulation**. The optical simulation of OLED devices was performed by using SETFOS 4.6 simulation program, in which the corresponding refractive index (n) and extinction coefficient (k) of each layer, dipole orientation factor of emitter, thickness of each layer and emission spectrum of EML were input. The optical constants of all the organic layers were measured using ellipsometer (M-2000, J. A. Woollam). The dipole orientation factor of emitter was set to be isotropic. During the optical simulation, the thicknesses of all layers were fixed, except the thickness of ETL was varied from 10 to 60 nm.

## Data availability

The data that support the plots within the paper are available from the corresponding author upon reasonable request.

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

## Acknowledgements

This work was supported financially by the Program for Building Regional Innovation Ecosystems of the Ministry of Education, Culture, Sports, Science and Technology, Japan and JST ERATO Grant Number JPMJER1305, Japan and Kyulux Inc. The computation was mainly carried out using the computer facilities at Research Institute for Information Technology, Kyushu University. The authors also acknowledge Ms. Nozomi Nakamura and Ms. Keiko Kusuhara for their technical assistance with this research.

## Author contributions

C.A. initiated and supervised the project. C.-Y.C. designed, synthesized and characterized the blue TADF emitters. C.-Y.C. performed the computational calculation, photophysical and electrochemical measurements of the TADF emitters. C.-Y.C. and M.T. fabricated the OLEDs and measured the device performance and stability. C.-Y.C. and C.A. contributed to the manuscript writing. M.T., H.N. and C.A. provided suggestions on experiments and writing manuscript. All authors discussed the progress of the research and reviewed the manuscript.

## Additional information

**Competing interests:** C.A. is the external advisor of one of the sponsors of this work (Kyulux). The remaining authors declare no competing interests.

