## [Peer Review File · Nature Communications]

Reviewers' comments:

Reviewer #1 (Remarks to the Author):

This manuscript describes two newly developed TADF emitters with sky-blue emission color and high stability in OLED device. One of the sky-blue TADF OLED with LT90 of 38 h outperforms the existing blue high-efficiency OLEDs, including blue PhOLEDs and TADF OLEDs. The improved device performance from device A to E is very impressive. However, there is no description of the molecular design and the reasons behind the dramatic stability enhancement (about 40 times) of 3Ph2CzCzBN compared to 4CzBN. I believe these additional discussions will be valuable to improve the scientific readability of the article. In short, I recommend this article to be published in Nature Communication after few minor revisions indicated below.

1. A recently article about improving device operational lifetime in Science Advances (DOI: 10.1126/sciadv.aa06910) was published by the authors. In this article, a similar strategy for enhancing device stability by using diphenylcarbazole (Ph2Cz) as the donor groups. Could the authors provide some reasons for enhancing the lifetime via the molecular modification.
2. Following the above question, I am curious about the molecule 4Ph2CzBN with four Ph2Cz groups. Could the authors provide some comments on the synthesis of the molecule and the device stability using the material as the dopant? It would be a helpful information for the TADF OLED society.
3. The value Tg in Table 1 should be Td (5% weight loss).
4. The material purity for fabricating good lifetime device is especially important. Indeed, the NMR spectra cannot accurately measure the purity of materials. Can the authors provide a concrete purity information of the two emitters, such as HPLC data.

Reviewer #2 (Remarks to the Author):

The authors appropriately revised the manuscript according to the reviewers' comments. I would recommend the acceptance of this manuscript after addressing the following suggestions, which are beneficial to make this work clearer to the readers. 1, the best comprehensive performance is from the device E, displaying a slightly blue shift in the device with reducing ETL thickness. The corresponding discussion should be presented in detail. 2, it is curious that two TADF emitters exhibit similar EL spectra without obvious shift in devices although six phenyl rings with extended conjugation are introduced. In addition, the emitter 3Ph2CzCzBN was synthesized in two steps and has three Ph2Cz groups rather than four ones. Did it mean that the asymmetrical structure were responsible for the stability?

We would like to take this opportunity to thank the reviewers for his/her valuable comments. In response to the reviewers' comments, we have made the following changes and responses (All changes have been highlighted in yellow):

Reviewer #1 (Remarks to the Author):

This manuscript describes two newly developed TADF emitters with sky-blue emission color and high stability in OLED device. One of the sky-blue TADF OLED with LT90 of 38 h outperforms the existing blue high-efficiency OLEDs, including blue PhOLEDs and TADF OLEDs. The improved device performance from device A to E is very impressive. However, there is no description of the molecular design and the reasons behind the dramatic stability enhancement (about 40 times) of 3Ph2CzCzBN compared to 4CzBN. I believe these additional discussions will be valuable to improve the scientific readability of the article. In short, I recommend this article to be published in Nature Communication after few minor revisions indicated below.

Q1. A recently article about improving device operational lifetime in Science Advances (DOI: 10.1126/sciadv.aao6910) was published by the authors. In this article, a similar strategy for enhancing device stability by using diphenylcarbazole (Ph2Cz) as the donor groups. Could the authors provide some reasons for enhancing the lifetime via the molecular modification.

A: We thank for the comment from this reviewer. The hetero-donor strategy for TADF emitters is very important to enhance the device stability of the TADF OLED. It is found that the introduction of two different donor units in the TADF emitter is effective to have a good mixing of the lowest-lying charge transfer and higher-lying localized triplet excited states, i.e T_1 and T_n excited states, which facilitates a faster reverse intersystem crossing rate (k_{RISC}). Eventually, a shorter delayed lifetime of the TADF emitter results in a decrease of the triplet exciton intensity, hence less hot triplet excitons. We added such design strategy in the molecular design part of the manuscript to have a clear description for the readers. On the other hand, in the present manuscript, asymmetrical and hetero-donor TADF emitter (**3Ph2CzCzBN**) was synthesized. One of the reasons for having asymmetrical molecule is the easier purification of the intermediate.

Q2. Following the above question, I am curious about the molecule 4Ph₂CzBN with four Ph₂Cz groups. Could the authors provide some comments on the synthesis of the molecule and the device stability using the material as the dopant? It would be a helpful information for the TADF OLED society.

A: Thank you for the valuable comment from the reviewer. We tried to synthesize **4Ph₂CzBN** for a comparison; however, the molecular weight of **4Ph₂CzBN** is too high (~ 1400), which is difficult to be sublimed. Sublimation at a harsher condition (2×10^{-2} Pa, >450 °C), thermal decomposition starts to occur. Therefore, it is difficult to determine the device performance and stability based on **4Ph₂CzBN**. On the other hand, the emission of **4Ph₂CzBN** is expected to be more redshifted into the undesired green region. Device based on **4Ph₂CzBN** should result in a greenish-blue EL and a CIE_y > 0.4.

Q3. The value T_g in Table 1 should be T_d (5% weight loss).

A: We are sorry for the mistake. Correction was made accordingly.

Q4. The material purity for fabricating good lifetime device is especially important. Indeed, the NMR spectra cannot accurately measure the purity of materials. Can the authors provide a concrete purity information of the two emitters, such as HPLC data.

A: Thank you for the comment by the reviewer. We attached the purity check by HPLC measurement in Supplementary Figs. S8 and S9.

Reviewer #2 (Remarks to the Author):

The authors appropriately revised the manuscript according to the reviewers' comments. I would recommend the acceptance of this manuscript after addressing the following suggestions, which are beneficial to make this work clearer to the readers. 1, the best comprehensive performance is from the device E, displaying a slightly blue shift in the device with reducing ETL thickness. The corresponding discussion should be presented in detail. 2, it is curious that two TADF emitters exhibit similar EL spectra without obvious shift in devices although six phenyl rings with extended conjugation are introduced. In addition, the emitter 3Ph₂CzCzBN was synthesized in two steps and has three Ph₂Cz groups rather than four ones. Did it mean that the asymmetrical structure were responsible for the stability?

Q1. The best comprehensive performance is from the device E, displaying a slightly blue shift in the device with reducing ETL thickness. The corresponding discussion should be presented in detail.

A: We appreciate the comment by the reviewer. We did do an optical simulation before, to estimate the effect of the thickness of the ETL (10 to 60 nm) on the EL emission spectrum based on device **D**. It was found that the EL emission would be blue-shifted with a thinner ETL layer. Therefore, we chose 20 nm-thick ETL (device **E**) as a trial, which indeed showed better CIE coordinates than that of device **D**. In addition to better CIE coordinates, device **E** also showed comparable device performance and stability to device **D** as well. A comprehensive discussion on device **E** and the corresponding optical simulation have been added in the manuscript and Supplementary Fig. S36, respectively.

Q2. It is curious that two TADF emitters exhibit similar EL spectra without obvious shift in devices although six phenyl rings with extended conjugation are introduced. In addition, the emitter 3Ph₂CzCzBN was synthesized in two steps and has three Ph₂Cz groups rather than four ones. Did it mean that the asymmetrical structure were responsible for the stability?

A: Thank you for the comments by the reviewer. In fact, **3Ph₂CzCzBN** showed a much red-shifted emission than that of **4CzBN** in solution. However, comparing to solution, **3Ph₂CzCzBN** showed a smaller stokes shift in doped film, which could be attributed to a more bulky molecular structure of **3Ph₂CzCzBN**. Eventually, the EL spectra of both compounds are similar. Asymmetrical structure and hetero-donor strategy of the TADF

emitter are important for enhancing the device stability. Since the introduction of the two different donor units would promote a good mixing of the lowest-lying charge transfer and higher-lying localized triplet excited states, i.e., T_1 and T_n excited states, which facilitates a faster reverse intersystem crossing rate (k_{RISC}). A faster k_{RISC} finally results in a short delayed lifetime of the TADF emitter. A short delayed lifetime of blue TADF emitter is highly desired, which is also true for a blue phosphorescent emitter with a short triplet decay lifetime. The short lifetimes of the triplet excitons either in TADF emitters or phosphors may reduce the formation of hot triplet excitons, thus avoid device degradation and enhance device stability. We revised the molecular design section of the manuscript accordingly.

REVIEWERS' COMMENTS:

Reviewer #1 (Remarks to the Author):

The authors have well addressed to my comments and provided the additional data requested. The material design strategy is valuable for general readers. I recommend that the manuscript is accepted by Nature Communications after the authors take care of the following minor point.

The various PLOQY statements in the main text are clear and distinct. However, the summarized values and their conditions in Table 1 are confused. Please add all PLOQYs (solution, degas and film) to Table 1 and address them clearly.

Reviewer #2 (Remarks to the Author):

According to authors' response and text, I think the manuscript is now suitable for publication in the journal.

We would like to take this opportunity to thank the reviewers for his/her valuable comments. In response to the reviewers' comments, we have made the following changes and responses (All changes have been highlighted in yellow):

Reviewer #1 (Remarks to the Author):

The authors have well addressed to my comments and provided the additional data requested. The material design strategy is valuable for general readers. I recommend that the manuscript is accepted by Nature Communications after the authors take care of the following minor point.

Q1 The various PLQY statements in the main text are clear and distinct. However, the summarized values and their conditions in Table 1 are confused. Please add all PLQYs (solution, degas and film) to Table 1 and address them clearly.

A: We are sorry for the confusing Table 1. We added all PLQYs and their corresponding conditions (solution, degassed, aerated and film) to Table 1.

Reviewer #2 (Remarks to the Author):

According to authors' response and text, I think the manuscript is now suitable for publication in the journal.

A: We thank the reviewer again for his/her comments.